# SET TRANSFORMER

## ABSTRACT

Many machine learning tasks such as multiple instance learning, 3D shape recognition and few-shot image classification are defined on *sets of instances*. Since solutions to such problems do not depend on the permutation of elements of the set, models used to address them should be *permutation invariant*. We present an attention-based neural network module, the *Set Transformer*, specifically designed to model interactions among elements in the input set. The model consists of an encoder and a decoder, both of which rely on attention mechanisms. In an effort to reduce computational complexity, we introduce an attention scheme inspired by *inducing point* methods from sparse Gaussian process literature. It reduces computation time of self-attention from quadratic to linear in the number of elements in the set. We show that our model is theoretically attractive and we evaluate it on a range of tasks, demonstrating increased performance compared to recent methods for set-structured data.

## 1 INTRODUCTION

Learning representations has proven to be an essential problem for deep learning and its many success stories. The majority of problems tackled by deep learning are *instance-based* and take the form of mapping a fixed-dimensional input tensor to its corresponding target value (Krizhevsky et al., 2012; Graves et al., 2013). For some applications, we are required to process *set-structured data*. Multiple instance learning (Dietterich et al., 1997; Maron & Lozano-Pérez, 1998) is an example of such a *set-input* problem, where a set of instances is given as an input and the corresponding target is a label for the entire set. Other problems such as 3D shape recognition (Wu et al., 2015; Shi et al., 2015; Su et al., 2015; Charles et al., 2017), sequence ordering (Vinyals et al., 2016), and various set operations (Muandet et al., 2012; Oliva et al., 2013; Edwards & Storkey, 2017; Zaheer et al., 2017) can also be viewed as such set-input problems. Moreover, many meta-learning (Thrun & Pratt, 1998; Schmidhuber, 1987) problems which learn using a set of different but related tasks may also be treated as set-input tasks where an input set corresponds to the training dataset of a single task. For example, few-shot image classification (Finn et al., 2017; Snell et al., 2017; Lee & Choi, 2018) operates by building a classifier using a *support set* of images, which is evaluated with query images.

A model for *set-input* problems should satisfy two critical requirements. First, it should be *permutation invariant* — the output of the model should not change under any permutation of the elements in the input set. Second, such a model should be able to process input sets of any size. While these requirements stem from the definition of a set, they are not easily satisfied in neursal-network-based models: classical feed-forward neural networks violate both requirements, and RNNs are sensitive to input order.

Recently, Edwards & Storkey (2017) and Zaheer et al. (2017) propose neural network architectures which meet both criteria, which we call *set pooling* methods. In this model, each element in a set is first independently fed into a feed-forward neural network that takes fixed-size inputs. Resulting feature-space embeddings are then aggregated using a *pooling* operation (mean, sum, max or similar). The final output is obtained by further non-linear processing of the aggregated embedding. This remarkably simple architecture satisfies both aforementioned requirements, and more importantly, is proven to be a universal approximator for any set function (Zaheer et al., 2017). Thanks to this property, it is possible to learn complex mapping between input sets and their target outputs in a black-box fashion, much like with feed-forward or recurrent neural networks.

Even though this set pooling approach is theoretically attractive, it remains unclear whether we can approximate complex mappings well using only instance-based feature extractors and simple pooling operations. Since every element in a set is processed independently in a set pooling operation, some information regarding interactions between elements has to be necessarily discarded. This can make some classes of problems unnecessarily difficult to solve.

Consider the problem of meta-clustering: we would like to learn a parametric mapping from an input set of points to centers of any clusters in the set, for many such sets. Even though a neural network with a set pooling operation can approximate such a mapping by learning to quantize space, this quantization cannot depend on the contents of the set. It limits the quality of the solution on one hand, and may make optimization of such a model more difficult; we show empirically in Section 4 that it leads to under-fitting.

In this paper, we propose a novel set-input deep neural network architecture called the *Set Transformer*, (*cf. Transformer*, Vaswani et al. (2017)). The novelty of the Set Transformer comes from three important design choices: 1) We use a self-attention mechanism based on the Transformer to process every element in an input set, which allows our approach to naturally encode pairwise- or higher-order interactions between elements in the set. 2) We propose a method to reduce the $O(n^2)$ computation time of Transformers to $O(nm)$ where $m$ is a fixed hyperparameter. 3) We use a self-attention mechanism to aggregate features, which is especially beneficial when the problem of interest requires multiple dependent outputs, such as the problem of meta-clustering, where the meaning of each cluster center heavily depends its location relative to the other clusters. We apply the Set Transformer to several set-input problems and empirically demonstrate the importance and effectiveness of these design choices.

This paper is organized as follows. In Section 2, we briefly review the concept of set functions, existing architectures, and the self-attention mechanism. In Section 3, we introduce Set Transformers, our novel neural network architecture for set functions. In Section 4, we present various experiments that demonstrate the benefits of the Set Transformer. We discuss related works in Section 5 and conclude the paper in Section 6.

## 2 BACKGROUND

### 2.1 POOLING ARCHITECTURE FOR SETS

Problems involving a set of objects have the *permutation invariance* property: the target value for a given set is the same regardless of the order of objects in the set. A simple example of a permutation invariant model is a network that performs pooling over embeddings extracted from the elements of a set. More formally,

$$\text{net}(\{x_1, \cdots, x_n\}) = \rho(\text{pool}(\{\phi(x_1), \cdots, \phi(x_n)\})). \tag{1}$$

Zaheer et al. (2017) has proven that all permutation invariant functions can be represented as (1) when pool is the sum operator and $\rho, \phi$ any continuous functions, thus justifying the use of this architecture for set-input problems.

Note that we can deconstruct (1) into two parts: an *encoder* ($\phi$) which independently acts on each element of a set of $n$ items, and a *decoder* ($\rho(\text{pool}(\cdot))$) which aggregates these encoded features and produces our desired output. Most network architectures for set-structured data follow this encoder-decoder structure. Our proposed method is also composed of an encoder and a decoder, but our embedding function $\phi$ does not act independently on each item but considers the whole set to obtain the embedding. Additionally, instead of a fixed function such as mean, our aggregating function $\text{pool}(\cdot)$ is parameterized and can thus adapt to the problem at hand.

### 2.2 ATTENTION

Assume we have $n$ query vectors (corresponding to $n$ points in an input set) each with dimension $d_q$: $Q \in \mathbb{R}^{n \times d_q}$. An attention function $\text{Att}(Q, K, V)$ is a function that maps queries $Q$ to outputs using $n_v$ key-value pairs $K \in \mathbb{R}^{n_v \times d_q}, V \in \mathbb{R}^{n_v \times d_v}$.

$$\text{Att}(Q, K, V; \omega) = \omega\left(QK^\top\right) V. \tag{2}$$

Table 1: Time complexity of various set operations. $n$ is the number of items, $d$ is the dimensionality of each item, and $m$ is the number of inducing points.

| Set operations | Time complexity | High-order interactions | Permutation invariant |
|---|---|---|---|
| Recurrent | $O(nd)$ | Yes | No |
| Pooling (Zaheer et al., 2017) | $O(nd)$ | No | Yes |
| Relational Networks (Santoro et al., 2017) | $O(n^2d)$ | Yes | Yes |
| Set Transformer (SAB + PMA, ours) | $O(n^2d)$ | Yes | Yes |
| Set Transformer (ISAB + PMA, ours) | $O(nmd)$ | Yes | Yes |

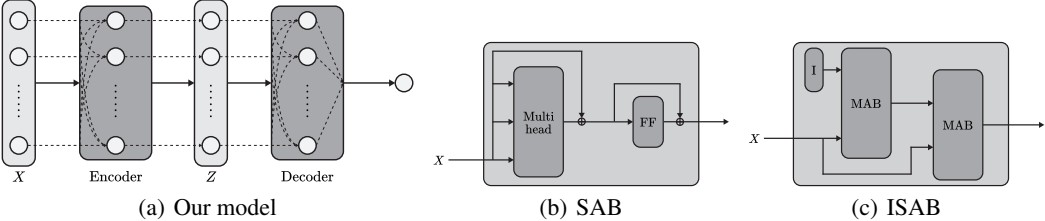

(a) Our model $\qquad$ (b) SAB $\qquad$ (c) ISAB

Figure 1: Diagrams of our attention-based set operations.

The pairwise dot product $QK^\top \in \mathbb{R}^{n \times n_v}$ measures how similar each pair of query and key vectors is, with weights computed with an activation function $\omega$. The output $\omega(QK^\top)V$ is a weighted sum of $V$ where a value gets more weight if its corresponding key has larger dot product with the query.

*Multi-head attention*, originally introduced in Vaswani et al. (2017), is an extension of the previous attention scheme. Instead of computing a single attention function, this method first projects $Q, K, V$ onto $h$ different $d_q^M, d_q^M, d_v^M$-dimensional vectors, respectively. An attention function $(\text{Att}(\cdot; \omega_j))$ is applied to each of these $h$ projections. The output is a linear transformation of the concatenation of all attention outputs:

$$\text{Multihead}(Q, K, V; \lambda, \omega) = \text{concat}(O_1, \ldots, O_h)W^O, \tag{3}$$

$$\text{where } O_j = \text{Att}(QW_j^Q, KW_j^K, VW_j^V; \omega_j) \tag{4}$$

Note that $\text{Multihead}(\cdot, \cdot, \cdot; \lambda)$ has learnable parameters $\lambda = \{W_j^Q, W_j^K, W_j^V\}_{j=1}^h$, where $W_j^Q, W_j^K \in \mathbb{R}^{d_q \times d_q^M}$, $W_j^V \in \mathbb{R}^{d_v \times d_v^M}$, $W^O \in \mathbb{R}^{hd_v^M \times d}$. A typical choice for the dimension hyperparameters is $d_q^M = d_q/h$, $d_v^M = d_v/h$, $d = d_q$. For brevity, we set $d_q = d_v = d$ and $d_q^M = d_v^M = d/h$ throughout the rest of the paper. Unless specified otherwise, we use the scaled softmax $\omega_j(\cdot) = \text{softmax}(\cdot/\sqrt{d})$, which our experiments showed worked robustly in most settings.

## 3 SET TRANSFORMER

In this section, we motivate and describe the *Set Transformer*: an attention-based neural network architecture that is designed to process sets of data. A Set Transformer consists of an encoder followed by a decoder (*cf.* Section 2.1). The encoder transforms a set of instances into a set of features, which the decoder transforms into the desired fixed-dimensional output.

### 3.1 ATTENTION-BASED SET OPERATIONS

We begin by defining our attention-based set operations. While existing pooling methods for sets obtain instance features independently of other instances, we use self-attention to concurrently encode the whole set. This gives the Set Transformer the ability to preserve pairwise as well as higher-order interactions among instances during the encoding process. For this purpose, we adapt the multihead attention mechanism used in Transformer. We emphasize that all blocks introduced here are neural network blocks with their own parameters, and not fixed functions.

Given matrices $X, Y \in \mathbb{R}^{n \times d}$ which represent two sets of $d$-dimensional vectors, we define the Multihead Attention Block (MAB) with parameters $\lambda$ as follows:

$$\text{MAB}(X, Y) = \text{LayerNorm}(H + \text{rFF}(H)), \tag{5}$$

$$\text{where} \quad H = \text{LayerNorm}(X + \text{Multihead}(X, Y, Y; \omega)), \tag{6}$$

where rFF is any row-wise feedforward layer (i.e. it processes each instance independently and identically), and LayerNorm is layer normalization (Ba et al., 2016). The MAB is an adaptation of the encoder block of the Transformer (Vaswani et al., 2017) without positional encoding and dropout. Using the MAB, we define the Set Attention Block (SAB) as

$$\text{SAB}(X) := \text{MAB}(X, X). \tag{7}$$

In other words, an SAB takes a set and performs self-attention between the elements in the set, resulting in a set of equal size. Since the output of SAB contains information about pairwise interactions between the elements in the input set $X$, we can stack multiple SABs to encode higher order interactions. Note that while the SAB (7) involves a multihead attention operation (6), where $Q = K = V = X$, it could reduce to applying a residual block on $X$. In practice, it learns more complicated functions due to linear projections of $X$ inside attention heads, (2) and (4).

A potential problem with using SABs for set-structured data is the quadratic time complexity $O(n^2)$, which may be too expensive for large sets ($n \gg 1$). We thus introduce the *Induced Set Attention Block* (ISAB), which bypasses this problem. Along with the set $X \in \mathbb{R}^{n \times d}$, additionally define $m$ $d$-dimensional vectors $I \in \mathbb{R}^{m \times d}$, which we call *inducing points*. Inducing points $I$ are part of the ISAB itself, and they are *trainable parameters* which we train along with other parameters of the network. An ISAB with $m$ inducing points $I$ is defined as:

$$\text{ISAB}_m(X) = \text{MAB}(X, H) \in \mathbb{R}^{n \times d}, \tag{8}$$

$$\text{where} \quad H = \text{MAB}(I, X) \in \mathbb{R}^{m \times d}. \tag{9}$$

The ISAB first transforms $I$ into $H$ by attending to the input set. The set of transformed inducing points $H$, which contains information about the input set $X$, is again attended to by the input set $X$ to finally produce a set of $n$ elements. This is analogous to low-rank projection or autoencoder models, where inputs ($X$) are first projected onto a low-dimensional object ($H$) and then reconstructed to produce outputs. The difference is that the goal of these methods is reconstruction whereas ISAB aims to obtain good features for the final task. We expect the learned inducing points to encode some global structure which helps explain the inputs $X$. As an example, think of a clustering problem on a 2D plane. The inducing points could be appropriately distributed points on the 2D plane so that the encoder can compare elements in the query dataset indirectly through their proximity to these grid points.

Note that in (8) and (9), attention was computed between a set of size $m$ and a set of size $n$. Therefore, the time complexity of $\text{ISAB}_m(X; \lambda)$ is $O(nm)$ where $m$ is a hyperparameter — an improvement over the quadratic complexity of the SAB. We compare characteristics of various set operations in Table 1. We also emphasize that both of our set operations are *permutation equivariant*:

**Definition 1.** *We say a function $f : X^n \to Y^n$ is permutation equivariant iff for any permutation $\pi \in S_n$, $f(\pi x) = \pi f(x)$. Here $S_n$ is the set of all permutations of indices $\{1, \cdots, n\}$.*

**Property 1.** *Both $\text{SAB}(X)$ and $\text{ISAB}_m(X)$ are permutation equivariant.*

## 3.2 ENCODER

Using the SAB and ISAB defined above, we construct the encoder $\text{Encoder} : X \mapsto Z \in \mathbb{R}^{n \times d}$ of the Set Transformer by stacking multiple SABs or multiple ISABs, for example:

$$\text{Encoder}(X) = \text{SAB}(\text{SAB}(X)) \tag{10}$$

$$\text{Encoder}(X) = \text{ISAB}_m(\text{ISAB}_m(X)). \tag{11}$$

We point out again that the time complexity for $\ell$ stacks of SABs and ISABs are $O(\ell n^2)$ and $O(\ell nm)$, respectively. This can result in much lower processing times when using ISAB (as compared to SAB), while still maintaining high representational power.

### 3.3 DECODER

After the encoder transforms data $X \in \mathbb{R}^{n \times d_x}$ into features $Z \in \mathbb{R}^{n \times d}$, the decoder aggregates them into a single vector which is fed into a feed-forward network to get final outputs. A common aggregation scheme is to simply take the average or dimension-wise maximum of the feature vectors (*cf.* Section 1). We instead aggregate features by applying multihead attention on a learnable set of $k$ seed vectors $S \in \mathbb{R}^{k \times d}$. We call this scheme *Pooling by Multihead Attention* (PMA):

$$\text{Decoder}(Z; \lambda) = \text{rFF}(\text{SAB}(\text{PMA}_k(Z))) \in \mathbb{R}^{k \times d} \qquad (12)$$

$$\text{where } \text{PMA}_k(Z) = \text{MAB}(S, \text{rFF}(Z)) \in \mathbb{R}^{k \times d}, \qquad (13)$$

Note that the output of $\text{PMA}_k$ is a set of $k$ items. In most cases, using one seed vector ($k = 1$) and no SAB sufficed. However, when the problem of interest requires $k$ correlated outputs, the natural thing to do is to use $k$ inducing points. An example of such a problem is clustering where the desired output is $k$ centers. In this case, the additional SAB was crucial because it allowed the network to directly take the correlation between the $k$ pooled features into account. Intuitively, feature aggregation using attention should be beneficial because the influence of each instance on the target is not necessarily equal. For example, consider a problem where the target value is the maximum value of a set of real numbers. Since the target can be recovered using only a single instance (the largest), finding and attending to that instance during aggregation will be advantageous. In the next subsection, we further analyze both the encoder and decoder structures more rigorously.

### 3.4 ANALYSIS

Since the blocks used to construct the encoder (i.e., SAB, ISAB) are permutation equivariant, the mapping of the encoder $X \to Z$ is permutation equivariant as well. Combined with the fact that the PMA in the decoder is a permutation invariant transformation, we have the following:

**Proposition 1.** *The Set Transformer is permutation invariant.*

Being able to approximate any function is a desirable property, especially for black-box models such as deep neural networks. Building on previous results about the universal approximation of permutation invariant functions, we prove the universality of Set Transformers:

**Proposition 2.** *The Set Transformer is a universal approximator of permutation invariant functions.*

*Proof.* See Appendix A. □

## 4 EXPERIMENTS

To evaluate the Set Transformer, we apply it to a suite of tasks involving sets of data points. We repeat all experiments five times and report performance metrics evaluated on corresponding test datasets. Along with baselines, we compared various architectures arising from the combination of the choices of having attention in encoders and decoders. Unless specified otherwise, "simple pooling" means average pooling.

- rFF + Pooling ( Zaheer et al. (2017)): rFF layers in encoder and simple pooling + rFF layers in decoder.

- rFFp-mean/rFFp-max + Pooling (Zaheer et al. (2017)): rFF layers with permutation equivariant variants in encoder (Eq (4) in Zaheer et al. (2017)) and simple pooling + rFF layers in decoder.

- rFF + Dotprod (Yang et al. (2018); Ilse et al. (2018)): rFF layers in encoder and dot product attention based pooling + rFF layers in decoder.

- SAB(ISAB) + Pooling: Stack of SABs (ISABs) in encoder and simple pooling + rFF layers in decoder.

- rFF + PMA: rFF layers in encoder and PMA (followed by stack of SABs) in decoder.

- Set Transformer: Stack of SABs (ISABs) in encoder and PMA (followed by stack of SABs) in decoder.

Table 2: Mean absolute errors on the max regression task.

| Architecture | MAE |
|---|---|
| rFF + Pooling (mean) | $2.133 \pm 0.190$ |
| rFF + Pooling (sum) | $1.902 \pm 0.137$ |
| rFF + Pooling (max) | $\mathbf{0.1355 \pm 0.0074}$ |
| Set Transformer | $0.2085 \pm 0.0127$ |

Table 3: Error rates on the unique character counting task.

| Architecture | Error |
|---|---|
| rFF + Pooling | $0.5618 \pm 0.0072$ |
| rFFp-mean + Pooling | $0.5383 \pm 0.0076$ |
| rFFp-max + Pooling | $0.5641 \pm 0.0077$ |
| rFF + Dotprod | $0.5529 \pm 0.0076$ |
| rFF + PMA | $0.5428 \pm 0.0076$ |
| SAB + Pooling | $0.4477 \pm 0.0077$ |
| Set Transformer | $\mathbf{0.4178 \pm 0.0075}$ |

## 4.1 TOY PROBLEM: MAXIMUM VALUE REGRESSION

To demonstrate the advantage of attention-based set aggregation over simple pooling operations, we consider a toy problem: regression to the maximum value of a given set. Given a set of real numbers $\{x_1, \cdots, x_n\}$, the goal is to return $\max(x_1, \cdots, x_n)$. Given prediction $p$, we use the mean absolute error $|p - \max(x_1, \cdots, x_n)|$ as the loss function. We constructed simple pooling architectures with three different pooling operations: $\max$, $\mathrm{mean}$, and $\mathrm{sum}$. We report loss values after training in Table 2. Mean- and sum-pooling architectures result in a high mean absolute error (MAE). The model with max-pooling can predict the output perfectly by learning its encoder to be an identity function, and thus achieves the highest performance. Notably, the Set Transformer achieves performance comparable to the max-pooling model, which underlines the importance of additional flexibility granted by attention mechanisms — it can learn to find and attend to the maximum element.

## 4.2 COUNTING UNIQUE CHARACTERS

In order to test the ability of modelling interactions between objects in a set, we introduce a new task of counting unique elements in an input set. We use the Omniglot (Lake et al., 2015) dataset, which consists of 1,623 different handwritten characters from various alphabets, where each character is represented by 20 different images.

We split all characters (and corresponding images) into train, validation, and test sets and only train using images from the train character classes. We generate input sets by sampling between 6 and 10 images and we train the model to predict the number of different characters inside the set. We used a Poisson regression model to predict this number, with the rate $\lambda$ given as the output of a neural network. We maximized the log likelihood of this model using stochastic gradient ascent.

We evaluated model performance using sets of images sampled from the test set of characters. Table 3 reports accuracy, measured as the frequency at which the mode of the Poisson distribution chosen by the network is equal to the number of characters inside the input set.

## 4.3 SOLVING MAXIMUM LIKELIHOOD PROBLS FOR MIXTURE OF GAUSSIANS

We applied the set-input networks to the task of maximum likelihood of mixture of Gaussians (MoGs). The log-likelihood of a dataset $X = \{x_1, \ldots, x_n\}$ generated from an MoG with $k$ components is

$$\log p(X; \theta) = \log p(X; \pi, \{\mu_j, \sigma_j\}_{j=1}^k) = \sum_{i=1}^n \log \sum_{j=1}^k \pi_j \mathcal{N}(x_i; \mu_j, \mathrm{diag}(\sigma_j^2)). \quad (14)$$

The goal is to learn the optimal parameters $\theta^*(X) = \arg\max_\theta \log p(X; \theta)$. The typical approach to this problem is to run an iterative algorithm such as Expectation-Maximisation (EM) until convergence. Instead, we aim to learn a generic meta-algorithm that directly maps the input set $X$ to $\theta^*(X)$. One can also view this as amortized maximum likelihood learning. Specifically, given a dataset $X$, we train a neural network to output parameters $f(X; \lambda) = \{\pi(X), \{\mu_j(X), \sigma_j(X)\}_{j=1}^k\}$ which maximize

$$\mathbb{E}_X \left[ \sum_{i=1}^{|X|} \log \sum_{j=1}^k \pi_j(X) \mathcal{N}(x_i; \mu_j(X), \mathrm{diag}(\sigma_j^2(X))) \right]. \quad (15)$$

We structured $f(\cdot; \lambda)$ as a set-input neural network and learned its parameters $\lambda$ using stochastic gradient ascent, where we approximate gradients using minibatches of *datasets*.

We tested Set Transformers along with other set-input networks on two types of datasets. We used four seed vectors for the PMA ($S \in \mathbb{R}^{4 \times d}$), the same as the number of clusters.

**Synthetic 2D mixtures of Gaussians**: Each dataset contains $n \in [100, 500]$ points on a 2D plane, each sampled from one of four Gaussians.

**CIFAR-100 meta-clustering**: Each dataset contains $n \in [100, 500]$ images sampled from four random classes in the CIFAR-100 dataset. Each image is represented by a 512-dim vector obtained from a pretrained VGG net (Simonyan & Zisserman, 2014).

Table 4: Meta clustering results. The number inside parenthesis indicates the number of inducing points used in ISABs of encoders. We show average likelihood per data for the synthetic dataset and the adjusted rand index (ARI) for the CIFAR-100 experiment. LL1/data, ARI1 are the evaluation metrics after a single EM update step. The oracle for the synthetic dataset is the log likelihood of the actual parameters used to generate the set, and the CIFAR oracle was computed by running EM until convergence.

|  | Synthetic | | CIFAR-100 | |
| --- | --- | --- | --- | --- |
| Architecture | LL0/data | LL1/data | ARI0 | ARI1 |
| Oracle | -1.4726 | | 0.9150 | |
| rFF + Pooling | $-2.0006 \pm 0.0123$ | $-1.6186 \pm 0.0042$ | $0.5593 \pm 0.0149$ | $0.5693 \pm 0.0171$ |
| rFFp-mean + Pooling | $-1.7606 \pm 0.0213$ | $-1.5191 \pm 0.0026$ | $0.5673 \pm 0.0053$ | $0.5798 \pm 0.0058$ |
| rFFp-max + Pooling | $-1.7692 \pm 0.0130$ | $-1.5103 \pm 0.0035$ | $0.5369 \pm 0.0154$ | $0.5536 \pm 0.0186$ |
| rFF + Dotprod | $-1.8549 \pm 0.0128$ | $-1.5621 \pm 0.0046$ | $0.5666 \pm 0.0221$ | $0.5763 \pm 0.0212$ |
| SAB + Pooling | $-1.6772 \pm 0.0066$ | $-1.5070 \pm 0.0115$ | $0.5831 \pm 0.0341$ | $0.5943 \pm 0.0337$ |
| ISAB (16) + Pooling | $-1.6955 \pm 0.0730$ | $-1.4742 \pm 0.0158$ | $0.5672 \pm 0.0124$ | $0.5805 \pm 0.0122$ |
| rFF + PMA | $-1.6680 \pm 0.0040$ | $-1.5409 \pm 0.0037$ | $0.7612 \pm 0.0237$ | $0.7670 \pm 0.0231$ |
| Set Transformer | $-1.5145 \pm 0.0046$ | $-1.4619 \pm 0.0048$ | $0.9015 \pm 0.0097$ | $0.9024 \pm 0.0097$ |
| Set Transformer (16) | $\mathbf{-1.5009 \pm 0.0068}$ | $\mathbf{-1.4530 \pm 0.0037}$ | $\mathbf{0.9210 \pm 0.0055}$ | $\mathbf{0.9223 \pm 0.0056}$ |

We report the performance of the oracle and of different models in Table 4. Additionally, it contains scores attained by all models after a single EM update. Overall, the Set Transformer found accurate parameters and even outperformed the oracles after a single EM update. This can be explained by relatively small size of the input sets, which leads to some clusters having fewer than 10 points. In this regime, sample statistics can differ from population statistics, which limits the performance of the oracle, but the Set Transformer can adapt accordingly. Notably, the Set Transformer with only 16 inducing points showed the best performance, even outperforming the full Set Transformer. We believe this is due to the knowledge transfer and regularization via inducing points, helping the network to learn global structures. Our results also imply that the improvements from using the PMA is more significant than that of using SAB, supporting our claim of the importance of attention-based decoders. We provide detailed generative processes, network architectures, and training schemes along with additional experiments with various numbers of inducing points in Appendix B.3.

## 4.4 META SET ANOMALY DETECTION

Table 5: Meta set anomaly results. Each architecture is evaluated using average of test area under receiver operating characteristic curve (AUROC) and test area under precision-recall curve (AUPR).

| Architecture | Test AUROC | Test AUPR |
| --- | --- | --- |
| Random guess | 0.5 | 0.125 |
| rFF + Pooling | $0.5643 \pm 0.0139$ | $0.4126 \pm 0.0108$ |
| rFFp-mean + Pooling | $0.5687 \pm 0.0061$ | $0.4125 \pm 0.0127$ |
| rFFp-max + Pooling | $0.5717 \pm 0.0117$ | $0.4135 \pm 0.0162$ |
| rFF + Dotprod | $0.5671 \pm 0.0139$ | $0.4155 \pm 0.0115$ |
| SAB + Pooling | $0.5757 \pm 0.0143$ | $0.4189 \pm 0.0167$ |
| rFF + PMA | $0.5756 \pm 0.0130$ | $0.4227 \pm 0.0127$ |
| Set Transformer | $\mathbf{0.5941 \pm 0.0170}$ | $\mathbf{0.4386 \pm 0.0089}$ |

We evaluate our methods on the task of meta-anomaly detection within a set using the CelebA dataset. The dataset consists of 202,599 images with the total of 40 attributes. We randomly sample

1,000 sets of images. For every set, we select two attributes at random and construct the set by selecting seven images containing both attributes and one image with neither. The goal of this task is to find the image that does not belong to the set. We give a detailed description of the experimental setup in Appendix B.4. Table 5 contains empirical results, which show that Set Transformers outperformed all other methods by a significant margin.

## 4.5 Point Cloud Classification

We evaluated Set Transformers on a classification task using the ModelNet40 (Chang et al., 2015) dataset, containing 40 categories of three-dimensional objects. Each object is represented as a point cloud, which we treat as a set of $n$ elements in $\mathbb{R}^3$. Table 6 contains experimental results on point clouds[1] with $n = 1000$ points each. In this setting, MABs turned out to be prohibitively expensive due to their $O(n^2)$ time complexity. Additional results with $n = 100$ points and experiment details are available in Appendix B.5. Note that ISAB (16) + Pooling outperformed Set Transformers (ISAB (16) + PMA (1)) by a large margin. Our interpretation is that the class of a point cloud object could be efficiently represented by simple aggregation of point features, and the PMA suffered from an optimization issue in this setting. We would like to point out that PMA outperformed simple pooling in all other experiments.

Table 6: Test accuracy for the point cloud classification task using 1,000 points.

| Architecture | Accuracy |
|---|---|
| rFF + Pooling | $0.8551 \pm 0.0142$ |
| rFF + PMA (1) | $0.8534 \pm 0.0152$ |
| ISAB (16) + Pooling | $\mathbf{0.8915 \pm 0.0144}$ |
| Set Transformer (16) | $0.8662 \pm 0.0149$ |
| rFF + Pooling (Zaheer et al., 2017) | $0.83 \pm 0.01$ |
| rFF + Pooling + tricks (Zaheer et al., 2017) | $0.87 \pm 0.01$ |

## 5 Related Works

**Pooling architectures for permutation invariant mappings** Pooling architectures for sets have been used in various problems such as 3D shape recognition (Shi et al., 2015; Su et al., 2015), discovering causality (Lopez-Paz et al., 2016), learning the statistics of a set (Edwards & Storkey, 2017), few-shot image classification (Snell et al., 2017), and conditional regression and classification (Garnelo et al., 2018). Zaheer et al. (2017) discusses the structure in general and provides a partial proof of the universality of the pooling architecture.

**Attention-based approaches for sets** Vinyals et al. (2016) proposes an architecture to map sets into sequences, where elements in a set are pooled by weighted average with weights computed from attention mechanism. Several recent works have highlighted the competency of attention mechanisms in modeling sets. (Yang et al., 2018) proposes AttSets for multi-view 3D reconstruction, where attention is applied to the encoded features of elements in sets before pooling. Similarly, (Ilse et al., 2018) uses an attention in pooling for multiple instance learning. However, the attention applied in those papers are simple dot-product attention that does not encode higher-order interactions between elements. Although not permutation invariant, (Mishra et al., 2018) has an attention as one of its core components to meta-learn to solve various tasks using sequences of inputs.

**Modeling interactions between elements in sets** An important reason to use the Transformer is to explicitly model higher-order interactions among the elements in a set. Santoro et al. (2017) proposes the relational network, a simple architecture that sum-pools all pairwise interactions of elements in a given set, but not higher-order interactions. Similarly to our work, Ma et al. (2018) uses the Transformer to model interactions between the objects in a video. They use mean-pooling to obtain aggregated features which they fed into an LSTM.

**Inducing point methods** The idea of letting trainable vectors $I$ directly interact with datapoints is loosely based on the inducing point methods used in sparse Gaussian processes (Quiñonero-Candela

---

[1]The point-cloud dataset used in this experiment was obtained directly from the authors of Zaheer et al. (2017).

& Rasmussen, 2005) and the Nyström method for matrix decomposition (Fowlkes et al., 2004). $m$ trainable inducing points can also be seen as $m$ independent memory cells accessed with an attention mechanism. The Differential Neural Dictionary (Pritzel et al., 2017) stores previous experience as key-value pairs and uses this to process queries. One can view the ISAB is the inversion of this idea, where queries $I$ are stored and the input features are used as key-value pairs.

## 6 CONCLUSION

In this paper, we introduced the Set Transformer, an attention-based set-input neural network architecture. Our proposed method uses attention mechanisms for both encoding and aggregating features, and we have empirically validated that both of them are necessary for modelling complicated interactions among elements of a set. We also proposed an inducing point method for self-attention, which makes our approach scalable to large sets. We also showed useful theoretical properties of our model, including the fact that it is a universal approximator for permutation invariant functions. To the best of our knowledge, no previous work has successfully trained a neural network to perform amortized clustering in a single forward pass. An interesting topic for future work would be to apply Set Transformers to meta-learning problems other than meta-clustering. In particular, using Set Transformers to meta-learn posterior inference in Bayesian models seems like a promising line of research. Another exciting extension of our work would be to model the uncertainty in set functions by injecting noise variables into Set Transformers in a principled way.

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

# Appendices

## A PROOFS

**Lemma 1.** *The mean operator* $\text{mean}(\{x_1, \cdots, x_n\}) = \frac{1}{n} \sum_{i=1}^{n} x_i$ *is a special case of dot-product attention with softmax.*

*Proof.* Let $s = \mathbf{0} \in \mathbb{R}^d$ and $X \in \mathbb{R}^{n \times d}$.

$$\text{Att}(s, X, X; \text{softmax}) = \text{softmax}\left(\frac{sX^\top}{\sqrt{d}}\right) X = \frac{1}{n} \sum_{i=1}^{n} x_i$$

□

**Lemma 2.** *The decoder of a Set Transformer, given enough nodes, can express any element-wise function of the form* $\left(\frac{1}{n} \sum_{i=1}^{n} z_i^p\right)^{\frac{1}{p}}$.

*Proof.* We first note that we can view the decoder as the composition of functions

$$\text{Decoder}(Z) = \text{rFF}(H) \tag{16}$$
$$\text{where} H = \text{rFF}(\text{MAB}(Z, \text{rFF}(Z))) \tag{17}$$

We focus on $H$ in equation (17). Since feed-forward networks are universal function approximators at the limit of infinite nodes, let the feed-forward layers in front and back of the MAB encode the element-wise functions $z \rightarrow z^p$ and $z \rightarrow z^{\frac{1}{p}}$, respectively. We let $h = d$, so the number of heads is the same as the dimensionality of the inputs, and each head is one-dimensional. Let the projection matrices in multi-head attention $(W_j^Q, W_j^K, W_j^V)$ represent projections onto the jth dimension and the output matrix $(W^O)$ the identity matrix. Since the mean operator is a special case of dot-product attention, by simple composition, we see that an MAB can express any dimension-wise function of the form

$$M_p(z_1, \cdots, z_n) = \left(\frac{1}{n} \sum_{i=1}^{n} z_i^p\right)^{\frac{1}{p}}. \tag{18}$$

□

**Lemma 3.** *A PMA, given enough nodes, can express sum pooling* $\left(\sum_{i=1}^{n} z_i\right)$.

*Proof.* We prove this by construction.

Set the seed $s$ to a zero vector and let $\omega(\cdot) = 1 + f(\cdot)$, where $f$ is any activation function such that $f(0) = 0$. The identiy, sigmoid, or relu functions are suitable choices for $f$. The output of the multihead attention is then simply a sum of the values, which is $Z$ in this case. □

We additionally have the following universality theorem for pooling architectures:

**Theorem 1.** *Models of the form* $\text{rFF}(\text{sum}(\text{rFF}(\cdot)))$ *are universal function approximators in the space of permutation invariant functions.*

*Proof.* See Appendix A of Zaheer et al. (2017). □

By Lemma 3, we know that $\text{decoder}(Z)$ can express any function of the form $\text{rFF}(\text{sum}(Z))$. Using this fact along with Theorem 1, we can prove the universality of Set Transformers:

**Proposition 2.** *The Set Transformer is a universal function approximator in the space of permutation invariant functions.*

*Proof.* By setting the matrix $W^O$ to a zero matrix in every SAB and ISAB, we can ignore all pairwise interaction terms in the encoder. Therefore, the $\text{encoder}(X)$ can express any instance-wise feed-forward network $(Z = \text{rFF}(X))$. Directly invoking Theorem 1 concludes this proof. □

While this proof required us to ignore the pairwise interaction terms inside the SABs and ISABs to prove that Set Transformers are universal function approximators, our experiments indicated that self-attention in the encoder was crucial for good performance.

## B    Experiment Details

In all implementations, we omit the feed-forward layer in the beginning of the decoder $(\mathrm{rFF}(Z))$ because the end of the previous block contains a feed-forward layer. All MABs (inside SAB, ISAB and PMA) use fully-connected layers with ReLU activations for rFF layers.

In the architecture descriptions, $\mathrm{FC}(d, f)$ denotes the fully-connected layer with $d$ units and activation function $f$. $\mathrm{SAB}(d, h)$ denotes the SAB with $d$ units and $h$ heads. $\mathrm{ISAB}_m(d, h)$ denotes the ISAB with $d$ units, $h$ heads and $m$ inducing points. $\mathrm{PMA}_k(d, h)$ denotes the PMA with $d$ units, $h$ heads and $k$ vectors. All MABs used in SAB and PMA uses FC layers with ReLU activations for FF layers.

### B.1    Max Regression

Given a set of real numbers $\{x_1, \cdots, x_n\}$, the goal of this task is to return the maximum value in the set $\max(x_1, \cdots, x_n)$. We construct training data as follows. We first sample a dataset size $n$ uniformly from the set of integers $\{1, \cdots, 10\}$. We then sample real numbers $x_i$ independently from the interval $[0, 100]$. Given the network's prediction $p$, we use the actual maximum value $\max(x_1, \cdots, x_n)$ to compute the mean absolute error $|p - \max(x_1, \cdots, x_n)|$. We don't explicitly consider splits of train and test data, since we sample a new set $\{x_1, \cdots, x_n\}$ at each time step.

Table 7: Detailed architectures used in the max regression experiments.

| Encoder | | Decoder | |
|---|---|---|---|
| FF | SAB | Pooling | PMA |
| $\mathrm{FC}(64, \mathrm{ReLU})$ | $\mathrm{SAB}(64, 4)$ | $\mathrm{mean}, \mathrm{sum}, \mathrm{max}$ | $\mathrm{PMA}_1(64, 4)$ |
| $\mathrm{FC}(64, \mathrm{ReLU})$ | $\mathrm{SAB}(64, 4)$ | $\mathrm{FC}(64, \mathrm{ReLU})$ | $\mathrm{FC}(1, -)$ |
| $\mathrm{FC}(64, \mathrm{ReLU})$ | | $\mathrm{FC}(1, -)$ | |
| $\mathrm{FC}(64, -)$ | | | |

We show the detailed architectures used for the experiments in Table 7. We trained all networks using the Adam optimizer (Kingma & Ba, 2015) with a constant learning rate of $10^{-3}$ and a batch size of 128 for 20000 batches, after which loss converged for all architectures.

### B.2    Counting Unique Characters

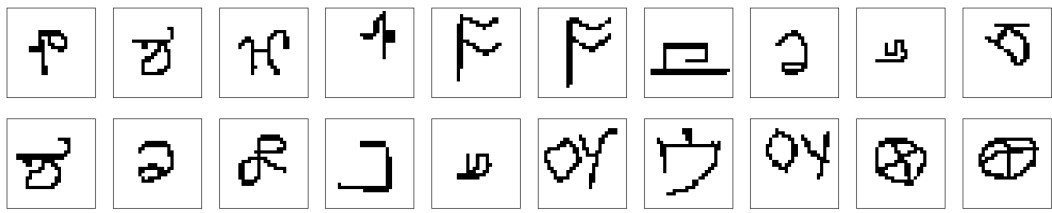

Figure 2: Try the task yourself: this is a randomly sampled set of 20 images from the Omniglot dataset. There are 14 different characters inside this set.

The task generation procedure is as follows. We first sample a set size $n$ uniformly from the set of integers $\{6, \cdots, 10\}$. We then sample the number of characters $c$ uniformly from $\{1, \cdots, n\}$. We sample $c$ characters from the training set of characters, and randomly sample instances of each character so that the total number of instances sums to $n$ and each set of characters has at least one instance in the resulting set.

We show the detailed architectures used for the experiments in Table 9. For both architectures, the resulting 1-dimensional output is passed through a softplus activation to produce the Poisson parameter $\gamma$. The role of softplus is to ensure that $\gamma$ is always positive.

Table 8: Detailed results for the unique character counting experiment.

| Architecture | Accuracy |
|---|---|
| rFF + Pooling | $0.4366 \pm 0.0071$ |
| rFF + PMA | $0.4617 \pm 0.0073$ |
| rFFp-mean + Pooling | $0.4617 \pm 0.0076$ |
| rFFp-max + Pooling | $0.4359 \pm 0.0077$ |
| rFF + Dotprod | $0.4471 \pm 0.0076$ |
| SAB + Pooling | $0.5659 \pm 0.0067$ |
| SAB + Dotprod | $0.5888 \pm 0.0072$ |
| Set Transformers (SAB + PMA (1)) | $\mathbf{0.6037 \pm 0.0072}$ |
| Set Transformers (SAB + PMA (2)) | $0.5806 \pm 0.0075$ |
| Set Transformers (SAB + PMA (4)) | $0.5945 \pm 0.0072$ |
| Set Transformers (SAB + PMA (8)) | $0.6001 \pm 0.0078$ |

Table 9: Detailed architectures used in the unique character counting experiments.

| Encoder | | Decoder | |
|---|---|---|---|
| rFF | SAB | Pooling | PMA |
| $\mathrm{Conv}(64, 3, 2, \mathrm{BN}, \mathrm{ReLU})$ | $\mathrm{Conv}(64, 3, 2, \mathrm{BN}, \mathrm{ReLU})$ | mean | $\mathrm{PMA}_1(8, 8)$ |
| $\mathrm{Conv}(64, 3, 2, \mathrm{BN}, \mathrm{ReLU})$ | $\mathrm{Conv}(64, 3, 2, \mathrm{BN}, \mathrm{ReLU})$ | $\mathrm{FC}(64, \mathrm{ReLU})$ | $\mathrm{FC}(1, \mathrm{softplus})$ |
| $\mathrm{Conv}(64, 3, 2, \mathrm{BN}, \mathrm{ReLU})$ | $\mathrm{Conv}(64, 3, 2, \mathrm{BN}, \mathrm{ReLU})$ | $\mathrm{FC}(1, \mathrm{softplus})$ | |
| $\mathrm{Conv}(64, 3, 2, \mathrm{BN}, \mathrm{ReLU})$ | $\mathrm{Conv}(64, 3, 2, \mathrm{BN}, \mathrm{ReLU})$ | | |
| $\mathrm{FC}(64, \mathrm{ReLU})$ | $\mathrm{SAB}(64, 4)$ | | |
| $\mathrm{FC}(64, \mathrm{ReLU})$ | $\mathrm{SAB}(64, 4)$ | | |
| $\mathrm{FC}(64, \mathrm{ReLU})$ | | | |
| $\mathrm{FC}(64, -)$ | | | |

The loss function we optimize, as previously mentioned, is the log likelihood $\log p(x|\gamma) = x \log(\gamma) - \gamma - \log(x!)$. We chose this loss function over mean squared error or mean absolute error because it seemed like the more logical choice when trying to make a real number match a target integer. Early experiments showed that directly optimizing for mean absolute error had roughly the same result as optimizing $\gamma$ in this way and measuring $|\gamma - x|$. We train using the Adam optimizer with a constant learning rate of $10^{-4}$ for $200,000$ batches each with batch size 32.

## B.3 SOLVING MAXIMUM LIKELIHOOD PROBLEMS FOR MIXTURE OF GAUSSIANS

### B.3.1 DETAILS FOR 2D SYNTHETIC MIXTURES OF GAUSSIANS EXPERIMENT

We generated the datasets according to the following generative process.

1. Generate the number of data points, $n \sim \mathrm{Unif}(100, 500)$.

2. Generate $k$ centers.
$$\mu_{j,d} \sim \mathrm{Unif}(-4, 4), \quad j = 1, \ldots, 4, \; d = 1, 2. \tag{19}$$

3. Generate cluster labels.
$$\pi \sim \mathrm{Dir}([1, 1]^\top), \quad z_i \sim \mathrm{Categorical}(\pi), \; i = 1, \ldots, n. \tag{20}$$

4. Generate data from spherical Gaussian.
$$x_i \sim \mathcal{N}(\mu_{z_i}, (0.3)^2 I). \tag{21}$$

Table 10 summarizes the architectures used for the experiments. For all architectures, at each training step, we generate 10 random datasets according to the above generative process, and updated the parameters via Adam optimizer with initial learning rate $10^{-3}$. We trained all the algorithms for $50k$ steps, and decayed the learning rate to $10^{-4}$ after $35k$ steps. Table 11 summarizes the detailed results with various number of inducing points in the ISAB. Figure 3 shows the actual clustering results based on the predicted parameters.

Table 10: Detailed architectures used in 2D synthetic experiments.

| Encoder | | | Decoder | |
|---|---|---|---|---|
| rFF | SAB | ISAB | Pooling | PMA |
| $FC(128, ReLU)$ | $SAB(128, 4)$ | $ISAB_m(128, 4)$ | mean | $PMA_4(128, 4)$ |
| $FC(128, ReLU)$ | $SAB(128, 4)$ | $ISAB_m(128, 4)$ | $FC(128, ReLU)$ | $SAB(128, 4)$ |
| $FC(128, ReLU)$ | | | $FC(128, ReLU)$ | $FC(4 \cdot (1 + 2 \cdot 2), -)$ |
| $FC(128, ReLU)$ | | | $FC(128, ReLU)$ | |
| | | | $FC(4 \cdot (1 + 2 \cdot 2), -)$ | |

Table 11: Average log-likelihood/data (LL0/data) and average log-likelihood/data after single EM iteration (LL1/data) the clustering experiment. The number inside parenthesis indicates the number of inducing points used in the SABs of encoder. For all PMAs, four seed vectors were used.

| Architecture | LL0/data | LL1/data |
|---|---|---|
| Oracle | -1.4726 | |
| rFF + Pooling | $-2.0006 \pm 0.0123$ | $-1.6186 \pm 0.0042$ |
| rFFp-mean + Pooling | $-1.7606 \pm 0.0213$ | $-1.5191 \pm 0.0026$ |
| rFFp-max + Pooling | $-1.7692 \pm 0.0130$ | $-1.5103 \pm 0.0035$ |
| rFF+Dotprod | $-1.8549 \pm 0.0128$ | $-1.5621 \pm 0.0046$ |
| SAB + Pooling | $-1.6772 \pm 0.0066$ | $-1.5070 \pm 0.0115$ |
| ISAB (16) + Pooling | $-1.6955 \pm 0.0730$ | $-1.4742 \pm 0.0158$ |
| ISAB (32) + Pooling | $-1.6353 \pm 0.0182$ | $-1.4681 \pm 0.0038$ |
| ISAB (64) + Pooling | $-1.6349 \pm 0.0429$ | $-1.4664 \pm 0.0080$ |
| rFF + PMA | $-1.6680 \pm 0.0040$ | $-1.5409 \pm 0.0037$ |
| Set Transformer | $-1.5145 \pm 0.0046$ | $-1.4619 \pm 0.0048$ |
| Set Transformer (16) | $-1.5009 \pm 0.0068$ | $-1.4530 \pm 0.0037$ |
| Set Transformer (32) | $\mathbf{-1.4963 \pm 0.0064}$ | $\mathbf{-1.4524 \pm 0.0044}$ |
| Set Transformer (64) | $-1.5042 \pm 0.0158$ | $-1.4535 \pm 0.0053$ |

### B.3.2 2D Synthetic Mixtures of Gaussians Experiment on Large-scale Data

To show the scalability of the set transformer, we conducted additional experiments on large-scale 2D synthetic clustering dataset. We generated the synthetic data as before, except that we sample the number of data points $n$ Unif(1000, 5000) and set $k = 6$. We report the clustering accuracy of a subset of comparing methods in Table 12. The set transformer with only 32 inducing points works extremely well, demonstrating its scalability and efficiency.

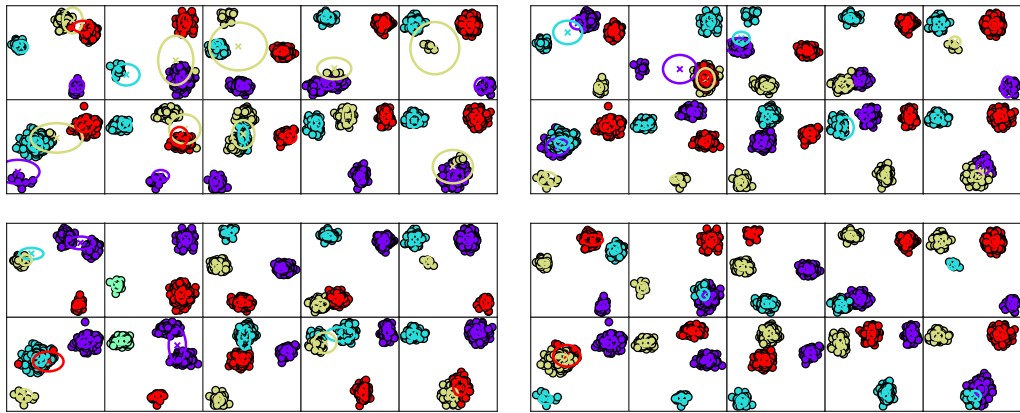

Figure 3: Clustering results for 10 test datasets, along with centers and covariance matrices. rFF+Pooling (top-left), SAB+Pooling (top-right), rFF+PMA (bottom-left), Set Transformer (bottom-right). Best viewed magnified in color.

Table 12: Average log-likelihood/data (LL0/data) and average log-likelihood/data after single EM iteration (LL1/data) the clustering experiment on large-scale data. The number inside parenthesis indicates the number of inducing points used in the SABs of encoder. For all PMAs, six seed vectors were used.

| Architecture | LL0/data | LL1/data |
|---|---|---|
| Oracle | -1.8202 | |
| rFF + Pooling | $-2.5195 \pm 0.0105$ | $-2.0709 \pm 0.0062$ |
| rFFp-mean + Pooling | $-2.3126 \pm 0.0154$ | $-1.9749 \pm 0.0062$ |
| rFF+PMA | $-2.0515 \pm 0.0067$ | $-1.9424 \pm 0.0047$ |
| Set Transformer(32) | $\mathbf{-1.8928 \pm 0.0076}$ | $\mathbf{-1.8549 \pm 0.0024}$ |

Table 13: Detailed architectures used in CIFAR-100 meta clustering experiments.

| Encoder | | | Decoder | |
|---|---|---|---|---|
| rFF | SAB | ISAB | rFF | PMA |
| $FC(256, ReLU)$ | $SAB(256, 4)$ | $ISAB_m(256, 4)$ | mean | $PMA_4(128, 4)$ |
| $FC(256, ReLU)$ | $SAB(256, 4)$ | $ISAB_m(256, 4)$ | $FC(256, ReLU)$ | $SAB(256, 4)$ |
| $FC(256, ReLU)$ | $SAB(256, 4)$ | $ISAB_m(256, 4)$ | $FC(256, ReLU)$ | $SAB(256, 4)$ |
| $FC(256, ReLU)$ | | | $FC(256, ReLU))$ | $FC(4 \cdot (1 + 2 \cdot 512), -)$ |
| $FC(256, ReLU)$ | | | $FC(256, ReLU)$ | |
| $FC(256, -)$ | | | $FC(256, ReLU)$ | |
| | | | $FC(4 \cdot (1 + 2 \cdot 512), -)$ | |

### B.3.3 DETAILS FOR CIFAR-100 META CLUSTERING EXPERIMENT

We pretrained VGG net (Simonyan & Zisserman, 2014) with CIFAR-100, and obtained the test accuracy 68.54%. Then, we extracted feature vectors of 50k training images of CIFAR-100 from the 512-dimensional hidden layers of the VGG net (the layer just before the last layer). Given these feature vectors, the generative process of datasets is as follows.

1. Generate the number of data points, $n \sim \text{Unif}(100, 500)$.

2. Uniformly sample four classes among 100 classes.

3. Uniformly sample $n$ data points among four sampled classes.

Table 13 summarizes the architectures used for the experiments. For all architectures, at each training step, we generate 10 random datasets according to the above generative process, and updated the parameters via Adam optimizer with initial learning rate $10^{-4}$. We trained all the algorithms for $50k$ steps, and decayed the learning rate to $10^{-5}$ after $35k$ steps. Table 14 summarizes the detailed results with various number of inducing points in the ISAB.

### B.4 META SET ANOMALY

Table 15 describes the architecture for meta set anomaly experiments. We trained all models via Adam optimizer with learning rate $10^{-4}$ and exponential decay of learning rate for 1,000 iterations. 1,000 datasets subsampled from CelebA dataset (see Figure 4) are used to train and test all the methods. We split 800 training datasets and 200 test datasets for the subsampled datasets.

### B.5 POINT CLOUD CLASSIFICATION

We used the ModelNet40 dataset for our point cloud classification experiments. This dataset consists of a 3-dimensional representation of 9,843 training and 2,468 test data which each belong to one of 40 object classes. As input to our architectures, we produce point clouds with $n = 100, 1,000, 5000$ points each (each point is represented by $(x, y, z)$ coordinates). For generalization, we randomly rotate and scale each set during training.

We show results our architectures in Table 16 and additional experiments which used $n = 100, 5000$ points in Table 17 and Table 18, respectively. We trained using the Adam optimizer with an initial learning rate of $10^{-3}$ which we decayed by a factor of 0.3 every $20,000$ steps. For the experiment

Table 14: Average clustering accuracies measured by Adjusted Rand Index (ARI) for CIFAR100 clustering experiments. The number inside parenthesis indicates the number of inducing points used in the SABs of encoder. For all PMAs, four seed vectors were used.

| Architecture | ARI0 | ARI1 |
|---|---|---|
| Oracle | 0.9151 | |
| rFF + Pooling | $0.5593 \pm 0.0149$ | $0.5693 \pm 0.0171$ |
| rFFp-mean + Pooling | $0.5673 \pm 0.0053$ | $0.5798 \pm 0.0058$ |
| rFFp-max + Pooling | $0.5369 \pm 0.0154$ | $0.5536 \pm 0.0186$ |
| rFF+Dotprod | $0.5666 \pm 0.0221$ | $0.5763 \pm 0.0212$ |
| SAB + Pooling | $0.5831 \pm 0.0341$ | $0.5943 \pm 0.0337$ |
| ISAB (16) + Pooling | $0.5672 \pm 0.0124$ | $0.5805 \pm 0.0122$ |
| ISAB (32) + Pooling | $0.5587 \pm 0.0104$ | $0.5700 \pm 0.0134$ |
| ISAB (64) + Pooling | $0.5586 \pm 0.0205$ | $0.5708 \pm 0.0183$ |
| rFF + PMA | $0.7612 \pm 0.0237$ | $0.7670 \pm 0.0231$ |
| Set Transformer | $0.9015 \pm 0.0097$ | $0.9024 \pm 0.0097$ |
| Set Transformer (16) | $\mathbf{0.9210 \pm 0.0055}$ | $\mathbf{0.9223 \pm 0.0056}$ |
| Set Transformer (32) | $0.9103 \pm 0.0061$ | $0.9119 \pm 0.0052$ |
| Set Transformer (64) | $0.9141 \pm 0.0040$ | $0.9153 \pm 0.0041$ |

Figure 4: Subsampled dataset examples. Each row is one dataset, which is composed of 7 normal images and 1 abnormal image (red box). Normal images in each subsampled dataset have both two attributes that are described in the rightmost column of figure. On the other hand, abnormal image does not contain the two attributes.

with 5000 points (Table 18), we increased the dimension of the attention blocks ($ISAB_{16}(512, 4)$ instead of $ISAB_{16}(128, 4)$) and also decayed the weights by a factor of $10^{-7}$. We also only used one ISAB block in the encoder because using two lead to overfitting in this setting.

## C  ADDITIONAL EXPERIMENTS

### C.1  RUNTIME OF SAB AND ISAB

We measured the runtime of SAB and ISAB on a simple benchmark (Figure 5). We used a single GPU (Tesla P40) for this experiment. The input data was a constant (zero) tensor of $n$ 3-dimensional vectors. We report the number of seconds it took to process 10000 sets of each size. The maximum set size we report for SAB is 2000 because the computation graph of bigger sets could not fit on our GPU. The specific attention blocks used are $ISAB_4(64, 8)$ and $SAB(64, 8)$.

Table 15: Detailed architectures used in CelebA meta set anomaly experiments. $\mathrm{Conv}(d, k, s, r, f)$ is a convolutional layer with $d$ output channels, $k$ kernel size, $s$ stride size, $r$ regularization method, and activation function $f$. If $d$ is a list, each element in the list is distributed. $\mathrm{FC}(d, f, r)$ denotes a fully-connected layer with $d$ units, activation function $f$ and $r$ regularization method. If $d$ is a list, each element in the list is distributed. $\mathrm{SAB}(d, h)$ denotes the SAB with $d$ units and $h$ heads. $\mathrm{PMA}(d, h, n_{\mathrm{seed}})$ denotes the PMA with $d$ units, $h$ heads and $n_{\mathrm{seed}}$ vectors. All MABs used in SAB and PMA uses FC layers with ReLU activations for rFF layers.

| Encoder | | Decoder | |
| --- | --- | --- | --- |
| rFF | SAB | Pooling | PMA |
| $\mathrm{Conv}([32, 64, 128], 3, 2, \mathrm{Dropout}, \mathrm{ReLU})$ | | mean | $\mathrm{PMA}_4(128, 4)$ |
| $\mathrm{FC}([1024, 512, 256], -, \mathrm{Dropout})$ | | $\mathrm{FC}(128, \mathrm{ReLU}, -)$ | $\mathrm{SAB}(128, 4)$ |
| $\mathrm{FC}(256, -, -)$ | | $\mathrm{FC}(128, \mathrm{ReLU}, -)$ | $\mathrm{FC}(256 \cdot 8, -, -)$ |
| $\mathrm{FC}([128, 128, 128], \mathrm{ReLU}, -)$ | $\mathrm{SAB}(128, 4)$ | $\mathrm{FC}(128, \mathrm{ReLU}, -)$ | |
| $\mathrm{FC}([128, 128, 128], \mathrm{ReLU}, -)$ | $\mathrm{SAB}(128, 4)$ | $\mathrm{FC}(256 \cdot 8, -, -)$ | |
| $\mathrm{FC}(128, \mathrm{ReLU}, -)$ | $\mathrm{SAB}(128, 4)$ | | |
| $\mathrm{FC}(128, -, -)$ | $\mathrm{SAB}(128, 4)$ | | |

Table 16: Detailed architectures used in the point cloud classification experiments.

| Encoder | | Decoder | |
| --- | --- | --- | --- |
| rFF | ISAB | Pooling | PMA |
| $\mathrm{FC}(256, \mathrm{ReLU})$ | $\mathrm{ISAB}(256, 4)$ | max | $\mathrm{Dropout}(0.5)$ |
| $\mathrm{FC}(256, \mathrm{ReLU})$ | $\mathrm{ISAB}(256, 4)$ | $\mathrm{Dropout}(0.5)$ | $\mathrm{PMA}_1(256, 4)$ |
| $\mathrm{FC}(256, \mathrm{ReLU})$ | | $\mathrm{FC}(256, \mathrm{ReLU})$ | $\mathrm{Dropout}(0.5)$ |
| $\mathrm{FC}(256, -)$ | | $\mathrm{Dropout}(0.5)$ | $\mathrm{FC}(40, -)$ |
| | | $\mathrm{FC}(40, -)$ | |

## C.2 NUMBER OF INDUCING POINTS AND PERFORMANCE

We trained $\mathrm{ISAB}_n + \mathrm{PMA}$ on the unique character counting task, varying the number of inducing points $n$. Accuracies are shown in Figure 6. Accuracies of other architectures (from Table 8) are shown as horizontal lines for comparison.

Note first that even the accuracy of $\mathrm{ISAB}_1 + PMA$ surpasses that of both rFF + Pooling and rFF + PMA. Also observe that accuracy tends to increase as $n$ increases. We additionally found that using larger values of $n$ (32, 64) did not increase accuracy by a significant margin. However, ISAB + PMA actually outperformed SAB + PMA in the clustering task (see Table 11).

Table 17: Additional point cloud experiments using 100 points.

| Architecture | Accuracy |
|---|---|
| rFF + Pooling | $0.7951 \pm 0.0166$ |
| rFF + PMA (1) | $0.8076 \pm 0.0160$ |
| ISAB (16) + Pooling | $0.8273 \pm 0.0159$ |
| Set Transformer (16) | $\mathbf{0.8454 \pm 0.0144}$ |
| rFF + Pooling + tricks (Zaheer et al., 2017) | $0.82 \pm 0.02$ |

Table 18: Additional point cloud experiments using 5000 points.

| Architecture | Accuracy |
|---|---|
| rFF + Pooling | $0.8933 \pm 0.0156$ |
| rFF + PMA (1) | $0.8628 \pm 0.0136$ |
| ISAB (16) + Pooling | $\mathbf{0.9040 \pm 0.0173}$ |
| Set Transformer (16) | $0.8779 \pm 0.0122$ |
| rFF + Pooling + tricks (Zaheer et al., 2017) | $\mathbf{0.90 \pm 0.003}$ |

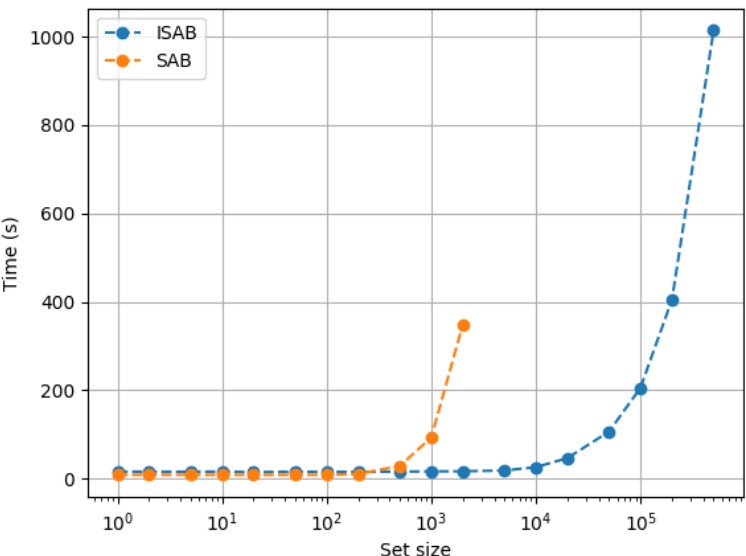

Figure 5: Runtime of a single SAB/ISAB block on dummy data. x axis is the size of the input set and y axis is time (seconds). Note that the x-axis is log-scale.

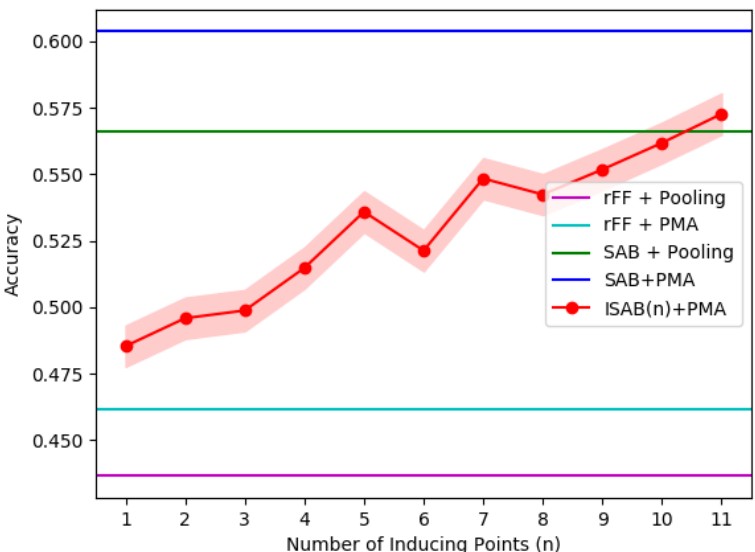

Figure 6: Accuracy of $\text{ISAB}_n + \text{PMA}$ on the unique character counting task. x-axis is $n$, the number of inducing points, and y-axis is accuracy.

