# OpenReview forum: "Set Transformer"
_ICLR.cc/2019/Conference_

### Official Review · AnonReviewer2 · 2018-11-02
**Missing comparisons to permutation equivariant DeepSets**

**Rating:** 6
**Confidence:** 5

**Review:**

This paper looks at stacking attention mechanism for learning over sets.

I think that the paper is well written overall. The architecture put forth is a fairly straightforward implementation of attention. Thus the methodological contribution is incremental. Still, it is nice to see some implementation of an attention model be considered for permutation invariant set embeddings.

However, there are some core misrepresentations and omissions that make publication difficult. The main problem is that the paper completely ignores the permutation equivariant mappings discussed in DeepSets (Zaheer 2017). See (4) and (23) of https://arxiv.org/pdf/1703.06114.pdf: "Since composition of permutation equivariant functions is also permutation equivariant, we can build deep models by stacking layers."
In practice this is often done by mapping points x_i in a set as x_i -> \phi(x_i) - max_j \phi(x_j). Stacking this layer works surprisingly well, typically better than just with a single pool. Thus, the permutation equivalent mappings of Zaheer 2017, which do have higher-order interactions and are linear in the number of points, are a glaring omission of table 1 and all of the experiments. Furthermore, the omission leads to a misrepresentation of the work.

Another unfortunate omission is previous work that considers set and distribution data through kernels and other nonparametric methods such as:
Muandet, Krikamol, et al. "Learning from distributions via support measure machines." Advances in neural information processing systems. 2012.
Oliva, Junier, Barnabás Póczos, and Jeff Schneider. "Distribution to distribution regression." International Conference on Machine Learning. 2013.

It is also odd that the paper compared to DeepSets on modelnet with 100 and 1000 points but not with 5000 points. Will there be code available?

Without a better description of and comparison to permutation equivariant mappings I would feel hesitant to recommend publication.

edit:
In light of the revised experiments and inclusion of permutation equivariant deepset layers, I'm inclined to recommend publication. However, if I could nitpick further, I think it would be nice to make some edit (or addition) to Table 1 to include permutation equivariant deepsets. Moreover, it would be nice to have some additional description of permutation equivariant layers in Section 2.1.

---

> ### Author Response · Authors · 2018-11-17
> **Added permutation equivariant baselines**
>
> Thanks for your constructive comments.
> i) Consider permutation equivariant mappings (Zaheer et al).
> Thanks for pointing this out. We added permutation equivariant architectures with both mean pooling and max pooling (rFFp-mean and rFFp-max) as baselines for all experiments, and have updated the paper. Our overall observation is that these permutation equivariant baselines do help, but the performance gain was not as significant as the gains achieved by SAB, ISAB and PMA.
>
> ii) Cite and consider Muandet et al. and Oliva et al.
> Thanks for mentioning the related works. Muandet et al. was cited and mentioned in the introduction in the submitted version of our paper. We have revised to include Oliva et al.
>
> iii) Add modelnet w/5000; will code be available?
> We had no time to conduct experiments with 5,000 pts during our first submission.
> Right now we are running experiments with 5,000 pts and they are going to be added to the appendix as soon as it is completed. The code will definitely be available open source.

---

> ### Author Response · Authors · 2018-11-30
> **Response after edits**
>
> Thank you very much for raising the score. Sorry for not updating Table 1, we were aware of it but forgot to update it when we uploaded our revision. We will correct it upon our acceptance. We will also try to discuss more about permutation equivariant layers as you suggested.

---

### Official Review · AnonReviewer1 · 2018-11-03
**Interesting paper that uses attention for set inputs but needs more ablation study**

**Rating:** 6
**Confidence:** 3

**Review:**

The paper proposes several variants of attention-based algorithms for set inputs. Compared with previous approach that processes each instance separately and then pooling, the proposed algorithm models the interactions among the instances within the set and performs better on tasks where such properties are important.

The experiments seem promising. The paper compares SAB and ISAB to rFF + pooling over multiple different tasks and SAB and ISAB outperform rFF + pooling in many tasks.

One drawback of the paper which limits its significance is that there are seemingly too many components and it is not clear which components are most important and which are not unnecessary. The authors can conduct some ablation study by removing some components and compare the performance to understand which parts are essential to the improvements.

---

> ### Author Response · Authors · 2018-11-17
> **About ablation studies**
>
> Thanks for your constructive comments.
> In our experiments, we compare (rFF+Pooling, SAB+Pooling, ISAB+Pooling, rFF+PMA, SAB+PMA, ISAB+PMA).
> Each of those variants are the Set Transformer with some (or no) components removed, so the experiments do report ablation results. We also added extra baselines (rFFp_mean + Pooling, rFFp_max + Pooling, rFF + Dotprod), and comparison to these methods supports our claim on the importance of having self-attention mechanism.

---

### Official Review · AnonReviewer3 · 2018-11-06
**A good paper but need some clarifications and improvements**

**Rating:** 5
**Confidence:** 4

**Review:**

This paper presented an attention-based neural network, namely set transformer, a new neural model
based on original transformer designed for set inputs. The basic idea is to introduce the attention
mechanism in both learning the feature embeddings of the set inputs during “encoding” and aggregating
these embeddings during “decoding”. The paper is written clearly and well motivated. The extensive
set of experiments were conducted to demonstrate the effectiveness of the proposed method. In general,
I like reading this paper but there are some limitations or unclear parts I need authors to clarify
and explain.

i) The proposed architecture is mainly adopted from the original transformer but it is highly related
to the baselines used in the experiments. For instance, it seems like that the current set
transformer is a simple combination of Yang et al.(2018) and Mishra et al.(2018) (using Stack of
SABs) in encoder side and of Ilse et al.(2018) (using PMA and stack of SABs) in the decoder side.
This simple combination makes the novelty of this paper unclear. I would like authors to clarify
more on the originality w.r.t. these previous works.

ii) Although authors proposed a variant of SABs - ISABs using landmark points to accelerate the
computation, there are no any runtime comparisons between SABs and ISABs by fixing other components.
It would be interesting to see that ISABs can approach the performance SABs and how it approaches it.
For instance, shall we expect that ISABs approach the performance of SABs when increasing the number
of landmark points (inducing points)? Since in practice most of datasets are relatedly large, I think
understanding the behavior of ISABs is a more interesting problem.

iii) After seeing the results in table 6, I have quite concerned about the practical performance of
set transformer on relatively large datasets (like 1000 points each class in the settings.) It looks
to me that not only set transformer may have computational issues to scale up, but more importantly
that when encoder learned really expressive embeddings with a relatively large number of the set
inputs it might be little need to leverage attention in pooling anymore. I would like authors to
conduct some other experiments on relatively large datasets to verify this hypothesis, which is
important for the practical applications of the proposed model.

---

> ### Author Response · Authors · 2018-11-17
> **Clarification for the novelty and additional experiments**
>
> Thanks for your constructive comments.
>
> i) Clarify originality
> Our method is not a simple combination of [1,2,3]. [1,3] uses dot product attention, where the transformed features are fed into a FF layer to produce softmax weights to be used to pool the features via weighted average. Hence, these methods do not take into account pairwise/higher-order interactions between elements in sets. We added dot-product attention based pooling as another baseline for all experiments. As we reviewed in the related works section, there are works using transformer-type self attention mechanism in encoder part of the model [2,4], but none of them were presented in context of permutation invariant set-taking neural nets. We summarize the novelty of our model below.
>
> - We adapted transformer based self-attention mechanism for *both* encoder and decoder part of permutation invariant set networks.
>
> - We introduce ISAB, which allows us to implement self-attention mechanism with reduced runtime complexity. This is an original contribution that was not present in previous works.
>
> - We introduce PMA, which differs from the dot-product attention-based pooling schemes presented in previous works. Especially, having multiple seed vectors and applying self-attention among them is a novel idea that we found to be very effective, especially for clustering-like problems, where modeling of output interactions (such as explaining away) is important.
>
> [1] Yang et al. 2018, Attentional aggregation of deep feature sets for multi-view 3d reconstruction.
> [2] Mishra et al. 2018, A simple neural attentive meta-learner.
> [3] Ilse et al. 2018, Attention-based deep multiple instance learning.
> [4] Ma et al. 2018, Attend and interact: higher-order object interactions for video understanding.
>
> ii) Runtime concerns; can Set Transformer scale up?
> ISABs should be able to scale up since they require O(n) memory and time, where n is the number of points in a set. In fact this is precisely why we introduced ISAB. We have added additional experiments to demonstrate actual running time of ISAB and SAB, and the tradeoff between accuracy and running time with respect to the different number of inducing points: see Appendix C.1 and Figure 5 in the revised paper.
>
> iii) Is attention useful when the set size is large and the embedding is expressive?
> First of all, please note that ISAB + Pooling is also our contribution, which performed the best in Table 6. We presume that the reason why the set transformer was not as effective as ISAB + Pooling in Table 6 was due to the nature of the problem. In point-cloud classification, once we encode interactions between elements via the self-attention mechanism, decoding them into label vectors does not require complex architectures like PMA. To verify this, we conducted extra experiments on clustering, where we used up to 5,000 data points per set. See Appendix B.3.2 and Table 12. In this experiment, where the PMA plays an important role, set transformer works extremely well with as few as 32 inducing points.

---

### Author Response · Authors · 2018-11-17
**Revision updated**

Dear reviewers,

Thanks for your comments. According to your opinion, we added three baselines to all experiments (mean pooling based permutation equivariant deep set , max pooling based permutation equivariant deep set (Zaheer et al, 2017), dot product attention based pooling (Yang et al., 2018, Ilse et al., 2018)). We've also added some extra experiments to see the scalability of the set transformer on large scale clustering experiments. Right now we are running the point cloud experiments with 5,000 pts, and the results will be updated as soon as it is completed.

There has been common concern about the novelty of our work. We want to emphasize again that our architecture is not a simple combination of existing works or naive adaptation of attention mechanism. Please refer to our comment to Reviewer 3 regarding the originality. Thanks.

---

### Meta-Review · Area_Chair1 · 2018-12-13
**Novelty is limited.**

**Confidence:** 4
**Recommendation:** Reject

**Metareview:**

This paper introduces set transformer for set inputs. The idea is built upon the transformer and introduces the attention mechanism. Major concerns on novelty were raised by the reviewers.